# Prenatal Nutrition Containing Bisphenol A Affects Placenta Glucose Transfer: Evidence in Rats and Human Trophoblast

**DOI:** 10.3390/nu12051375

**Published:** 2020-05-11

**Authors:** Linda Benincasa, Maurizio Mandalà, Luana Paulesu, Laura Barberio, Francesca Ietta

**Affiliations:** 1Department of Life Science, University of Siena, 53100 Siena, Italy; lindabeninc@gmail.com (L.B.); francesca.ietta@unisi.it (F.I.); 2Department of Biology, Ecology and Earth Sciences, University of Calabria, 87036 Rende, Italy; m.mandala@unical.it (M.M.); laura.barberio@unical.it (L.B.)

**Keywords:** food contaminants, pregnancy, placental glucose transfer, GLUT1, HTR-8/SVneo cells

## Abstract

This work aims to clarify the effect of dietary supplementation with Bisphenol A (BPA), a chemical widely present in beverage and food containers, on placental glucose transfer and pregnancy outcome. The study was performed on female Sprague Dawley rats fed with a diet containing BPA (2.5, 25 or 250 μg/Kg/day) for a period of a month (virgin state) plus 20 days during pregnancy. Western blot analysis and immunohistochemistry were performed in placental tissues for glucose type 1 transporter (GLUT1). Furthermore, human trophoblast, HTR8-SV/neo cells, were used to evaluate the effect of BPA on glucose transport and uptake. Studies in rats showed that food supplementation with BPA, produces a higher fetal weight (FW) to placenta weight (PW) ratio at the lowest BPA concentration. Such low concentrations also reduced maternal weight gain in late pregnancy and up-regulated placental expression of GLUT1. Treatment of HTR8-SV/neo with the non-toxic dose of 1 nM BPA confirmed up-regulation of GLUT1 expression and revealed higher activity of the transporter with an increase in glucose uptake and GLUT1 membrane translocation. Overall, these results indicate that prenatal exposure to BPA affects pregnancy and fetal growth producing changes in the placental nutrients-glucose transfer.

## 1. Introduction

The placenta is the interface between mother and fetus providing nutrients and gases to the fetus and removing waste products.

Glucose is the main metabolic fuel used by mammalian cells and is one of the major nutrients transferred from mother to fetus throughout gestation [1]. In early gestation, there is not a significant fetal gluconeogenesis activity [2], which means that physiological placental glucose homeostasis is intimately related to maternal supply.

The transfer of glucose across the human placenta is one-way directed to the fetus thanks to the establishment of a maternal-fetal gradient [3]. Glucose crosses placental membranes through a family of facilitated glucose transporters (the GLUT family) [4]. The glucose transporter 1 (GLUT1) is considered to be the primary placental transporter of glucose from the maternal circulation to the placenta [5]. GLUT1 plays an important role in the survival of the pre-implantation embryo [6] and throughout much of fetal development, and consequently homozygous GLUT1 null mice do not survive beyond embryonic day 14 [7]. GLUT1 is responsible for mediating placental transfer of glucose in humans and levels of placental GLUT1 are altered under various pathological conditions that affect the normal fetus growth [8].

Among these alterations, great concern is given to Bisphenol A (BPA), a synthetic phenol extensively used in the manufacture of polycarbonate plastics and epoxy resins. The main exposure to BPA occurs through beverage and food contact materials such as plastic food containers, food packaging, epoxy coating in metal cans and thermic paper [9,10,11].

BPA is one of the potential chemicals dangerous for prenatal human exposure, due to its presence in the amniotic fluid, placenta, and blood of the umbilical cord and its ability to cross the placenta and reach the fetus [12,13,14]. In addition, in vitro exposure to low-dose BPA is able to act at the maternal-fetal interface, altering trophoblast endocrine secretion and capability to migrate and invade maternal uterine tissues [15,16,17]. On the maternal side, cell decidualization as well as uterine receptivity to the implanting blastocyst, are both impaired by BPA [18].

Moreover, it is widely acknowledged that many neurological, cardiovascular and metabolic diseases could result from an unfavorable uterine environment that establishes following prenatal BPA exposure [19,20]. An interesting report by Rajakumar et al. 2015 showed that BPA augments glucose transporter-1 gene expression in primary human placental trophoblast cells [21]. However, in which way BPA is able to interfere with glucose uptake and how this interference reflects on an incorrect fetal development has to be elucidated.

This study aimed to investigate prenatal exposure to BPA and the associated glucose placental transport. As explained thereafter, the study was performed in female rats fed with a diet containing BPA at three different doses: 2.5, 25 and 250 μg/Kg of body weight/day. Human trophoblast, HTR-8/SVneo, cell cultures were also used to evaluate the effect of a direct exposure to BPA on placenta glucose transfer. The two models, albeit on different species, will help to highlight the effect of BPA by a maternal chronic exposure and an acute exposure directly on the placenta.

## 2. Materials and Methods

### 2.1. Animals

Experiments were performed on female Sprague Dawley rats at 8 weeks of age. Rats were divided into four groups treated with BPA at 2.5 μg/Kg/day BPA (2.5 BPA) (n = 7); 25 μg/Kg/day BPA (25 BPA) (n = 9); 250 μg/Kg/day of BPA (250 BPA) (n = 7), and with BPA vehicle, Ethanol, (Control BPA) (n = 7). The drugs BPA or ethanol were added in drinking water, on the base of daily water drink and the body weight, for a continue period of a month (virgin state) plus 20 days during pregnancy started from the 1st day of gestation, in rat the total gestation period is 22 days. Pregnant animals were obtained by placing a female in proestrus with a fertile male overnight, detection of spermatozoa using a vaginal smear on the following morning was used to confirm day 1 of pregnancy. All rats were housed individually in the Animal Care facility, maintained under controlled conditions on a 12 h light/dark photoperiod and provided commercial chow and tap water ad libitum. Animals, used for experimental purposes on the 20th day of pregnancy, were euthanized with isoflurane inhalation, followed by decapitation with a small animal guillotine, the uterus was removed and fetuses and placentas were taken. All experiments were conducted in accordance with the European Guidelines for the care and use of laboratory animals (Directive 26/2014/EU) and were approved by the local ethical committee of the University of Calabria and the Italian Ministry of Health (n.74/2018-PR).

### 2.2. Pregnancy Measurement and Tissue Processing

The mothers’ weight was checked each week during either the pre-pregnancy or pregnancy time, to evaluate the effect of BPA on the body weight after the total period of treatment in pre-pregnant and in pregnant state. Litter size, fetal weight, and placental weight from each of the three BPA groups (n = 6 with 2.5 μg/Kg/day; n = 8 with 25 μg/Kg/day; n = 7 with 250 μg/Kg/day) and the control group (n = 6) were checked at the time of animal sacrifice (20th day of pregnancy). All placentae and fetus were harvested from each rat. Two placentae were collected, always from the same position along the uterus independently of the gender; one was frozen at −80 °C for subsequent protein extraction and western blot analysis and one fixed in 10% buffered formalin then, embedded in paraffin for immunohistochemistry. The different rat number between the groups is because no all the rats used for the study mated.

### 2.3. Immunohistochemistry for Placental GLUT1

Five µm sections, from formalin fixed-paraffin embedded rat placenta tissues, were performed, making sure that a full face (including labyrinth and Spongio zone) were visible for each specimen. The sections were heated to 60 °C for 10 min, rehydrated and finally washed in Tris-buffered saline (TBS) (20 mM Tris-HCl and 150 mM NaCl, pH 7.6). Subsequently, the slides were subjected to antigenicity recovery in 10 mM citrate buffer at pH 6.0 in a microwave at 750 W (three cycles of 5 min) and washed in TBS. The slides were then pre-incubated with Protein Block Serum-Free (Dako, Santa Clara, CA, USA) to block the non-specific binding of antibodies and then incubated overnight at 4 °C with a polyclonal rabbit anti-GLUT1 antibody (Thermo Fisher Scientific, Waltham, MA, USA), diluted 1:200 in TBS. After washing, the slides were incubated with the appropriate anti-rabbit secondary antibody peroxidase-conjugated (Dako) diluted 1:500 in TBS and the reaction was detected by Diaminobenzidine (DAB) after counterstaining with Mayer’s Hematoxylin. In negative controls, the anti-GLUT1 antibody was replaced with TBS.

### 2.4. HTR-8/SVneo Cell Culture and Treatment

The HTR-8/SVneo cell line was originally generated from villous explants at early pregnancy and generously donated by Professor Charles Graham (Queens University, Kingston, Canada). The cells were cultured in Dulbecco’s Modified Eagle Medium (DMEM) medium without phenol red (Sigma Chemical Co, St. Louis, MO, USA) supplemented with 10% (*v*/*v*) fetal bovine serum (FBS) (Biochrom, Berlin, Germany), 100 U/mL penicillin/streptomycin, and 2 mM glutamine (Sigma Chemical Co) (complete medium) at 37 °C, in humidified 95% air-5% CO_2_. For each experiment, HTR-8/SVneo cells were seeded in 25 cm^2^ flasks, in 6 or 96 well plates, depending on the type of analyses, in complete medium. After cell adhesion, the medium was replaced with fresh DMEM containing 2% (*v*/*v*) FBS and added with 1 nM BPA (Sigma Chemical Co) or Ethanol (0.1%). Each treatment was done at least in triplicate. Cultures were further incubated up to 48 h, depending on the experiment, then assayed for cell proliferation, glucose uptake, GLUT1 expression and, GLUT1 membrane translocation (see specific sections).

### 2.5. Protein Extraction and Western Blot Analysis

Rat placental tissues and HTR-8/SVneo cell cultures were lysed in Mammalian Protein Extraction Reagent (MPER)^®^ lysis buffer (Thermo Fisher Scientific) with the addition of 1% (*v*/*v*) of Halt^TM^ Protease and Phosphatase Inhibitor Single-Use cocktail (Thermo Fisher Scientific). For Membrane and cytoplasm protein extraction, the Subcellular Protein Fractionation Kit for Cultured Cells (Thermo Fisher Scientific) was used. Protein content was determined through Bradford assay and 30 µg of total protein were loaded on 10% acrylamide/bis-acrylamide gel electrophoresis in condition of constant voltage 135 V per 75 min in TGS buffer (25 mM Tris, 190 mM glycine, 0.1%(*w*/*v*) SDS).

Proteins were electro-transferred onto nitrocellulose membranes and then blocked in 5% non-fat dry milk in Tris-buffered saline pH 7.2 (TBS) containing 0.1% Tween20 for 1 hour at room temperature. Membranes were then incubated with polyclonal rabbit anti-GLUT1 antibody (Thermo Fisher Scientific) or Sodium/potassium-transporting ATPase subunit alpha-1 (ATPA1) (Thermo Fisher Scientific) diluted 1:1000 in 3% *w*/*v* non-fat dry milk in TBS-Tween, and anti–human and –rat β-actin or GAPDH (Santa Cruz Biotechnology, Dallas, TX, USA) (at dilution of 1:3000 in 3% *w*/*v* non-fat dry milk in TBS-Tween) overnight at 4 °C. After that, membranes were washed and incubated for 1 h at room temperature with the appropriated horseradish peroxidase-conjugated IgG (BioRad, Hercules, CA, USA). The reaction was revealed using chemiluminescent reagent (BioRad) and the membranes digitalized with CHEMIDOC Quantity One 1D (BioRad).

### 2.6. Cell Viability

To evaluate the viability of HTR-8/SVneo cells following treatment with BPA, the Sulforhodamine B (SRB) assay was performed as described by Orellana and Kasinski [22]. Cells were seeded in 96-well plates at a density of 2.5 × 10^4^ cells/mL and incubated until the following day to allow adhesion. One plate culture was stopped at this time (time zero, T0), to show the initial number of cells. The other plates were treated with 1 nM BPA or equal amount of medium (Control) and incubated for further 48 h (T48). At both T0 and T48 plates, the medium was removed and after three times washing with PBS, cultures were fixed with 100 μL of 10% *v*/*v* ice-cold trichloroacetic acid (TCA) and incubated at 4 °C for 30 min. From this point, both T0 and T48 plates were processed simultaneously for SRB assay.

To obtain the percentage of cell proliferation the following formula was applied:$$\%\;\mathrm{cell}\;\mathrm{proliferation}=\frac{T48\;Absorbance}{T0\;Absorbance}\;\times \;100$$

### 2.7. Cell Glucose Uptake

5 × 10^3^ HTR-8/SVneo cells/well were seeded in 96 well plates, treated with 1 nM BPA or only medium (Control), and incubated for 48 h at 37 °C. Thereafter, cells were washed in PBS and dispensed with DMEM containing 10 mM glucose and 2%(*v*/*v*) FBS for 10 min at 37 °C, then the plate was placed on ice to stop cell metabolic activities.

Each well was incubated with a solution of cold PBS added of cold Inactivation Solution (0.6N HCl) for 5 min on a plate stirrer and Neutralization Solution (Tris Base 1M) on shake for 30–60 s. The intracellular glucose content was measured after incubation with Amplex Red Glucose Assay Kit^®^ (Thermo Fisher Scientific) for 30 min covered by light. The reaction was revealed by measuring the absorbance through a microplate reader in the range 530–560 nm. Data were expressed as percent of glucose uptake in BPA-treated versus control not-treated cells.

### 2.8. Immunofluorescence

5 × 10^3^ HTR-8/SVneo cells/well were seeded on rounded slides in six well plates and treated as described above for 48 h at 37 °C. After incubation, slides were washed and fixed for 10 min in ice-cold methanol. The slides were blocked in 5% Bovine Serum Albumin (BSA) for 30 min and incubated overnight with polyclonal rabbit anti-human GLUT1 antibody (Thermo Fisher Scientific) 1:100 in PBS and mouse anti-human ATP1A1 (Thermo Fisher) 1:200 in PBS. Slides were then washed in PBS and incubated 1 h with goat anti-rabbit IgG, DyLight^TM^ 550 (Thermo Fisher Scientific) diluted 1:1000 in PBS and goat anti-mouse IgG (H&L), DyLight^TM^ 488 (Thermo Fisher Scientific), for GLUT1 and ATPA1, respectively. The slides were counterstained with DAPI nuclear marker (Sigma Chemical Co) 1:1000 in PBS. Immunofluorescence was revealed with the confocal microscope and analyzed through LEICA LAS-X cellular imaging (LEICA Microsystem).

### 2.9. Data Analysis

All data sets from in vivo experiments were analyzed for normality using the D’Agostino Pearson-Omnibus test followed by either parametric (ANOVA following Dunnett’s test) or non-parametric tests (Kruskal-Wallis’test) as appropriate. The Spearman correlation coefficient was calculated to evaluate the correlation between the fetal weight and the placenta weight. Densitometric analysis from in vitro experiments was performed by an open source ImageJ software (NIH) and responses between BPA exposed and un-treated control cultures were compared by unpaired *t*-test. The bands of the blots were measured as mean density gray value for area. *p* values ≤ 0.05 were considered significant. Data are presented as means  ±  SEM or medians. Number of rats, samples, or experiments performed is indicated in the corresponding figure legends. GraphPad Prism 6.0 (GraphPad Software, Inc., San Diego, CA, USA) was used for the statistical analyses. Colocalization analysis were performed with the JACoP of ImageJ application (NIH). Colocalization was estimated by Pearson coefficient that measures the amount or degree of correlation of the mean intensities of the fluorophores.

## 3. Results

### 3.1. BPA Impact in Rats Fed with a Diet Containing BPA

#### 3.1.1. Maternal and Fetal Outcome

The mothers’ weight during the pre-pregnancy and the pregnancy state, placenta and fetal weight were measured at the time of sacrifice (Day 20) as well as the number of live offspring were measured (see Table 1 and Figure 1).

As reported in Table 1, non-pregnant rats showed a statistically significant increase in body weight in the second and the third week, in 2.5 and 25 BPA groups, compared to the previous week. No significant difference in weight was observed between any BPA group and control group at any week of treatment or in the total length of the period (four weeks) of BPA treatment. 250 BPA showed to be completely ineffective.

During pregnancy, the mother’s weight at the third week was significantly lower in 2.5 BPA group compared to control group. No significant difference was observed in the total weight gain for any BPA group as respect to control group nor in the litter size (Table 1).

Figure 1 shows that fetal weight at birth was significantly increased in rats treated with 2.5 BPA compared to control group (A) while placenta weight was reduced in 2.5 and 250 BPA fed rats (B).

To evaluate placental efficiency in BPA-treated animals, the ratio between fetal weight (FW) and placental weight (PW) was also examined. Figure 1C shows data on the 2.5 BPA group, the only one showing statistically significant results. FW/PW ratio was significantly higher in BPA group than in control group. FW and PW showed a positive and significant correlation in the two groups of animals (controls, *r* = 0.436, *r*^2^ = 0,19, *p* = 0.0002; 2.5 BPA, *r* = 0.553, *r*^2^ = 0,30, *p* < 0.0001) (Figure 1D). As shown, for a given FW corresponded a lower placenta weight in 2.5 BPA than in control rats.

#### 3.1.2. GLUT1 Placenta Expression and Localization

To investigate whether a higher placental efficiency with consequent increase in fetal birth weight could reflect a greater ability of the placenta to transport glucose, we examined placental GLUT1 expression and tissue localization. The western blot data reported in Figure 2A show a protein band of 45–55 kDa, corresponding to GLUT1 both in control and in the BPA groups. Densitometry analysis against β-actin revealed a statistically significant up-regulation of GLUT1 expression in rats fed with BPA 2.5 and 250 μg/Kg/day in comparison to control group (Figure 2B).

Immunohistochemistry was performed to evaluate GLUT1 localization in rat placenta of BPA-treated and control rats. Figure 3A,B reports data from the rats fed with BPA 2.5 μg/Kg/day the one showing more clear differences from the control group. Qualitative staining density for GLUT1, determined by researchers blinded to treatment, revealed a wider expression of GLUT1 in the decidual zone of BPA-treated than control rats. Higher magnification of the decidual-junctional zone revealed higher GLUT1 expression on the plasma membranes of trophoblast giant cells and in the spongiotrophoblast. At the fetal side (labyrinth zone), higher immunoreactivity for GLUT1 was shown in the chorionic projection with fetal blood capillaries and in the syncytiotrophoblast cells, surrounding the fetal vessels.

### 3.2. BPA Impact on HTR-8/SVneo Human Trophoblast Cells

#### 3.2.1. GLUT1 Cell Culture Expression

HTR-8/SVneo cell cultures were used to investigate the direct effect of BPA on GLUT1 in human trophoblast. These cells originated from human first trimester placenta and immortalized by transfection with a cDNA construct that encodes the simian virus 40 large T antigen [23]. Cultures were exposed for 48 h to 1 nM BPA, a usual concentration for in vitro studies [15] and considered to be environmentally relevant [12]. Western blot analysis showed that GLUT1 protein expression was significantly up-regulated in BPA-treated cells as respect to control untreated cells (Figure 4A,B).

#### 3.2.2. Cell Viability and Glucose Uptake

Cell viability, examined by the SRB assay, showed no toxicity of 1 nM BPA in our in vitro system (Figure 4C). To evaluate the effect of BPA on trophoblast glucose transport, glucose uptake by HTR-8/SVneo cells was examined. As shown in Figure 4D, glucose uptake was higher in BPA-treated than in control cells.

#### 3.2.3. GLUT1 Cell Membrane Translocation

To be active, the GLUT1 transporter must be translocated from the cytoplasm to the cell surface in contact with the external environment. Thus, we assessed GLUT1 membrane translocation through the co-localization of GLUT1 (red) with ATPA1 (green), a marker of plasma membrane, by immunofluorescence. As shown in Figure 5A, GLUT1/ATP1A1 co-localization showed by yellow staining, was observed on plasma membranes of contiguous cells with higher intensity and wider distribution in BPA-treated than in control untreated cultures. Colocalization was estimated by Pearson coefficient (Ct: r = 0.5; BPA: r = 0.80). Moreover, a stick of GLUT1 in perinuclear position, possibly corresponding to Golgi was detected in control and BPA-treated cultures (Figure 5A).

The immunoblot on HTR-8/SVneo subcellular fractioned protein extracts, respectively from cytoplasm and plasma membranes, confirmed a higher presence of GLUT1 in cell membranes of BPA-treated than control cultures (Figure 5B). Induction of GLUT1 membrane translocation by BPA was also supported by the higher Membrane/Cytoplasm (M/CP) ratio in BPA-treated than untreated cultures (Figure 5C).

## 4. Discussion

The main result of the present study is that prenatal nutrition containing Bisphenol A produces changes in placental glucose transporter 1 (GLUT1) and on a range of important pregnancy outcomes including fetal weight and the ratio of fetal weight (FW) to placenta weight (PW). The impact of BPA on placenta GLUT1 was supported by in vitro exposure on an immortalized human placental cell line, HTR-8/SVneo cells.

Among the three BPA concentrations used (2.5, 25, and 250 μg/Kg/day), the lowest (2.5, μg/Kg/day) was the most active, particularly on the increased expression of placental glucose transporter GLUT1 and in FW/PW ratio. Although at a lesser extent, 250 BPA was also active on induction of placental GLUT1 while 25 BPA was completely ineffective compared to untreated controls. The results obtained at different BPA concentrations support a dose-dependent activity of this chemical, with low doses being more effective than high doses [24].

The increase of FW/PW ratio in 2.5 BPA rats, suggests a higher efficiency of placenta in response to the low BPA dose. The FW/PW ratio indicates the amount of fetal tissue developed per gram of placenta, so, the higher the ratio, the greater the efficiency of the placenta [25].

The placenta is a plastic organ capable of morphological and functional adaptive changes to ensure appropriate fetal growth, particularly when it is compromised by adverse uterine conditions [26]. In this study, pregnancy was exposed to BPA, a harmful environmental chemical that easily reaches the fetus passing through the placenta, as also revealed by the concentrations detected in amniotic fluid, cord blood and fetal tissues [12].

Many studies were carried out to evaluate the influence of BPA on the weight at birth although the data are still inconsistent mainly because of the variety on models and methods of analysis used [27]. Our study in rats showed that low doses of BPA in in drinking water are correlated with an increase in fetal weight.

However, although a high birth weight on a low placental weight makes the FW/PW greater, suggesting a higher placental efficiency, this situation could represent an advantage in case of a normal weight fetus [28]. Otherwise, a too small or large fetus, demonstrates failure of the placental compensatory response [28]. Barker (1997; 1990) showed that size at birth is critical in determining not only neonatal viability but also an increased risk of diseases later in life, such as glucose intolerance and Type 2 diabetes [29,30].

It now remains to be asked whether the increase in placental efficiency leading to a higher fetal growth is the result of a direct effect of BPA on the placenta or an indirect effect by alteration of maternal metabolism which in turn causes functional adaptative functional changes in the placenta nutrient transfer. BPA is indeed largely known as an obesogenic agent contributing to insulin resistance and pancreatic β-cell dysfunction in pregnancy potentially playing a role in the pathogenesis of pregnancy complications, such as gestational diabetes mellitus [31]. Our study showed no change in the total gestational weight gain between BPA-treated and control rats. Changes were however detected in the weight of the mothers at day 20 of gestation, the time of sacrifice, with a significantly lower weight in 2.5 BPA rats than in control untreated rats.

These data suggest that changes in maternal metabolism, if any, might occur in BPA rats, in the last week of gestation. Unlike pregnancy, the weight of non-pregnant rats exposed to BPA was significantly increased in the second and third week of 2.5 and 25 BPA vs. control but not in 250 BPA, suggesting activity of lower BPA concentration in the metabolism of fertile rats.

Overall, our study indicates a differential impact of BPA on the metabolism between pregnancy and non-pregnant fertile status. However, further studies are needed to evaluate the effects of BPA in these different reproductive states. This is a particularly urgent issue because variations in both gestational and pre-pregnancy weight were linked to an increased risk of adverse maternal and fetus outcomes [32].

In mammals, intrauterine growth is mainly determined by the placental supply of nutrients to the fetus [33]. BPA can affect placentation as shown by in vitro and in vivo studies [16,34]. In a recent report, Müller et al. [34] showed that BPA exposure during early pregnancy affects uterine remodeling and results in growth restricted fetus.

Our study on placental tissues at approximately full term gestation, showed up-regulation of GLUT 1 expression in rats treated with BPA during the twenty days of pregnancy and for one month of the pre-gestational period.

When examining the different zones of the rat placenta, rats fed with 2.5 BPA showed an increase in GLUT1 expression, in terms of number of immunoreactive cells in the maternal decidual zone, mainly localizing in the plasma membranes of trophoblast giant cells and in the spongiotrophoblasts. In the labyrinth zone, the main site for nutrient transfer, higher immunoreactivity for GLUT1 was shown in the cells of the chorionic projection with fetal blood capillaries and in the syncytiotrophoblast cells, surrounding the fetal vessels. These results suggest an effect of BPA determining a plastic rearrangement of GLUT1 orientation in the maternal-placental layers in order to uptake from mother tissues as much glucose as possible to ensure an appropriate supply to the developing fetus.

Experiments on HTR-8/SVneo human trophoblast cells were performed to examine the effect of a direct exposure to BPA. These transformed trophoblast cell line, is representative of trophoblast invasiveness, a specific feature of fetal chorionic cells in the hemochorial-types placentas such as in humans and in rodents [23] HTR-8/SVneo experiments showed that BPA increases the expression and plasma membrane location of GLUT1 that in turn bring to higher glucose uptake from the cells. These results together with those obtained in rat placenta are in line with the report of Rajakumar et al. [21] showing an increase of GLUT1 mRNA levels in BPA-exposed human placental cells. How BPA triggers the translocation of GLUT1 is unknown. According to Perrini et al. [35], we can speculate that BPA can alter the PI3K pathway involved in GLUT1 translocation [35]. Future studies will focus on this aspect. It is important to underline that the effective dose of 1 nM BPA in trophoblast cells corresponds to the concentration present in tissues and fluids during pregnancy [12]. Previous studies showed that these same or even lesser concentrations of BPA alter the secretion of hormones as well as the invasive activity of human trophoblast [13,16]. The present studies revealed that such low BPA concentration is also capable of altering the transport of nutrients in trophoblast. The in vivo studies in rats fed with low doses of BPA indicated that the increase in placental GLUT1 leads to an increase in fetal weight, a consequence that might be harmful to the health of the fetus and his future life.

## 5. Conclusions

In conclusion, the data show clear evidence of the prenatal impact of BPA on placental glucose transport and fetal growth. The use of two types of approaches (in vivo in rats, in vitro in human trophoblast cells), showed superimposable effects, although with differences among species and placental histological structure.

BPA is ubiquitous in our daily life and every living being, including fertile or pregnant women, is exposed to this substance through contaminated food and water.

Given the importance of prenatal life for the health of the fetus and his future life, we hope that our studies, together with others on the same topic, help increase the perception of the harmful effects of BPA and encourage the implementation of lows to limit the use of BPA in products for women who wish to become mothers.

## Figures and Tables

**Figure 1 nutrients-12-01375-f001:**
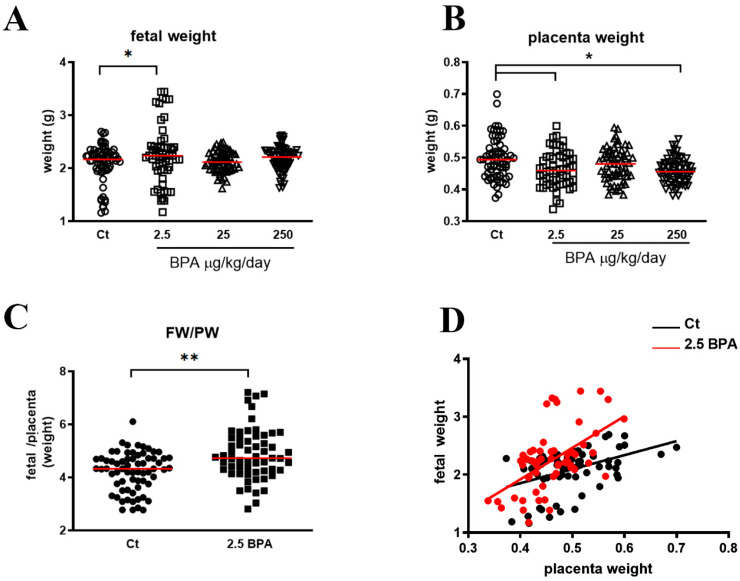
Maternal and fetal outcome. (**A**) Fetal weight in control (Ct) (n = 67), 2.5 BPA (n = 59), 25 BPA (n = 64) and 250 BPA (n = 68). (**B**) Placenta weight in control (Ct) (n = 67), 2.5 BPA (n = 59), 25 BPA (n = 64) and 250 BPA (n = 68). (**C**) Fetal weight and placenta weight ratio in control (Ct) and 2.5 BPA fed rats. (**D**) Correlation of fetal weight vs. placenta weight in control (Ct; black) (y = 2.426x + 0.8816; *r* = 0.436, *r*^2^ = 0.19, *p* = 0.0002) and 2.5 BPA (red) (y = 5.450x − 0.2609; *r* = 0.553, *r*^2^ = 0.30, *p* < 0.0001) fed rats. * *p* < 0.05; ** *p* < 0.01. Data are expressed as scatter plot and median (red line).

**Figure 2 nutrients-12-01375-f002:**
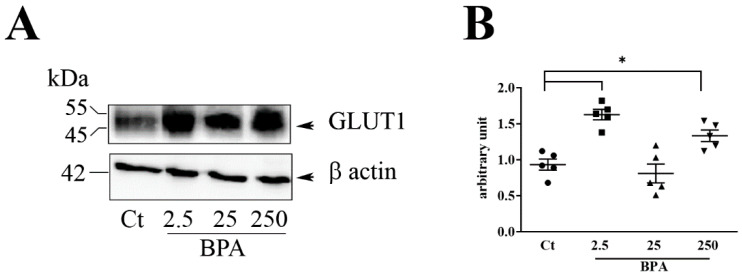
Effect of BPA on GLUT1 protein level in placentas from pregnant rats. (**A**) Representative immunoblot and (B) Densitometric analysis, performed in placentas collected from pregnant rats fed with different concentration of BPA (ug/Kg/day): 2.5, 25, 250 or Ethanol 0.1% (Control: Ct) (n  =  5 for each treatment). ANOVA * *p* < 0.05.

**Figure 3 nutrients-12-01375-f003:**
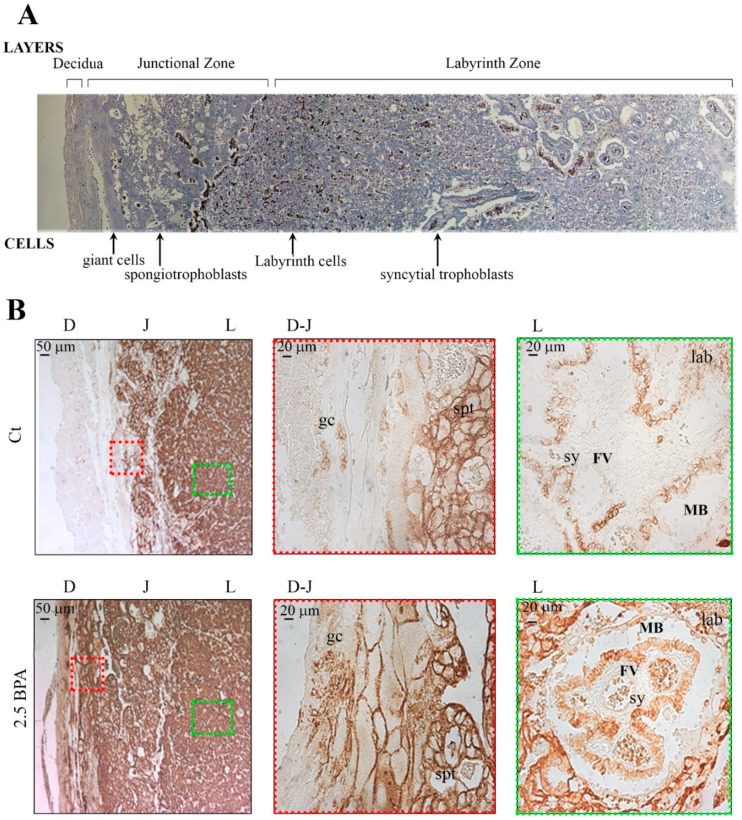
GLUT1 localization in placentas from pregnant rats exposed or not to BPA. (**A**) For orientation purposes: representative image montage of H&E staining of the rat maternal-fetal interface (gestational day 20) showing fetal trophoblast cells distribution in the different layers of the placenta. (**B**) Representative immunohistochemical analysis of GLUT1 localization at the maternal-fetal interface (gestational day 20). Brownish staining represents positive immunoreactivity for GLUT1 in control (Ct; upper panels) and 2.5 BPA (lower panels) fed rat mothers. On the left (upper and lower panels) Decidua (D), Junctional zone (J) and Labyrinth zone (L) at lower magnification (50×). Red square represents the selected area for the acquisition of the higher magnification (200×) for the transition zone Decidua-Junctional (D-J) showed in the middle. Green square represents the selected area for the acquisition of the higher magnification (200×) for the Labyrinth zone (L) showed on the right. gc = giant cells; spt = spongiotrophoblast cells, lab: labyrinth cells; sy: Syncytiotrophoblast; FV: Fetal vessel; MB: Maternal blood.

**Figure 4 nutrients-12-01375-f004:**
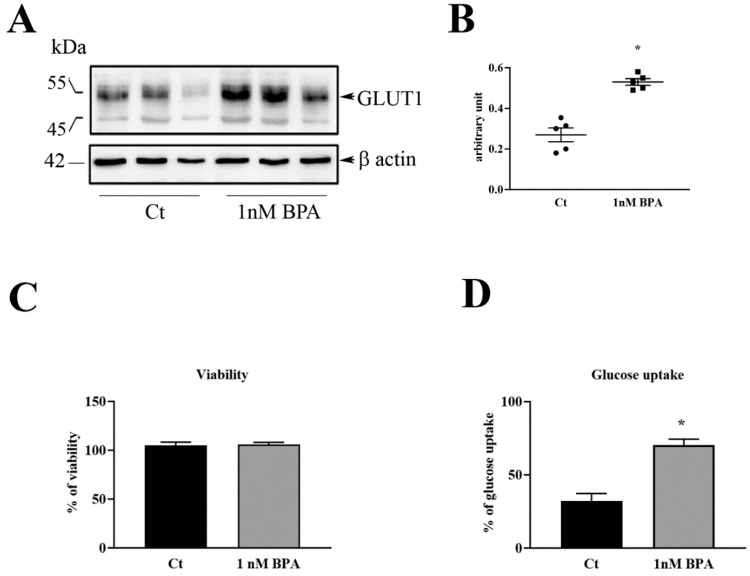
Effect of BPA on GLUT1 protein level, cell viability and glucose uptake in HTR-8/Svneo cells. (**A**) Representative immunoblot of three experiments for GLUT1 expression in HTR-8/Svneo cells exposed for 48 h to 1 nM BPA or only to medium (control: Ct). (**B**) Densitometric analysis (n  =  5 for each treatment). (**C**) Cell viability by SRB assay, in cells exposed for 48 h to 1 nM BPA or only medium (Control: Ct). Data are reported as percent of the initial (T0) number of cells. (**D**) Glucose uptake in cell cultures exposed for 48 h to 1 nM BPA or only medium (Control: Ct). *t*-test * *p* < 0.05.

**Figure 5 nutrients-12-01375-f005:**
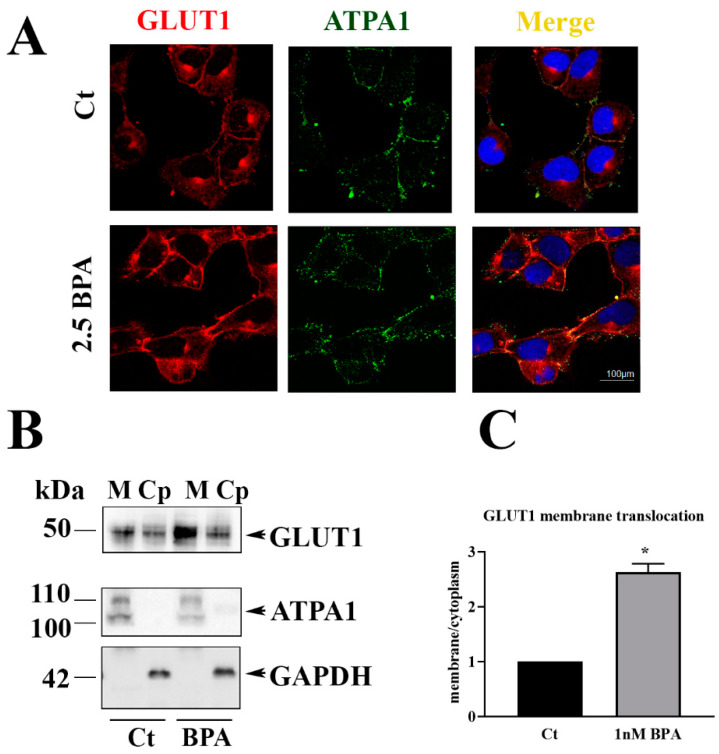
GLUT1 plasma membrane translocation in HTR-8/SVneo. (**A**) Representative confocal microscopy for GLUT1 (red) and ATPA1 (green) in HTR8/Svneo exposed to 1 nM BPA or only medium (control: Ct). GLUT1/ATP1A1 co-localization (yellow: merge). Colocalization was estimated by Pearson coefficient (Ct: r = 0.5; BPA: r = 0.80). (**B**) Representative immunoblot for GLUT1 in subcellular fractioned protein extracts, M: Plasma membranes; Cp: Cytoplasm. (**C**) Membrane/Cytoplasm ratio from densitometric analysis of three experiments. Cytoplasm fractions were normalized with GAPDH, membrane fraction for ATP1A1. Significance (* *p* < 0.05) was calculated by *t*-test.

**Table 1 nutrients-12-01375-t001:** Maternal weight changes during the pre-pregnancy and the pregnancy time.

	Control	BPA 2.5	BPA 25	BPA 250
No. of pregnant dams	7	7	9	7
Pre-Pregnancy Body Weight (g) (mean ± SEM)				
1st day	175.7 ± 18.0	143.1 ± 3.6	146.0 ± 3.8	148.8 ± 5.8
2nd week	197.9 ± 17.1	169.0 ± 3.7 ^#^	159.8 ± 1.6 ^#^	156.0 ± 9.4
3th week	209.9 ± 13.7	187.9 ± 2 °	182.4 ± 3.2 °	183.5 ± 4.5
4th week	216.6 ± 13.7	196.0 ± 3.3	192.4 ± 3.6	200.3 ± 6.0
Total weight change	48.7 ± 4.5	50.8 ± 6.6	41.7 ± 1.4	49.1 ± 4.6
No. of pregnant dams	7	6	8	7
Pregnancy body weight (g) (mean ± SEM)				
1st week	235 ± 15.2	222.7 ± 10.0	211.4 ± 2.8	220.5 ± 5.0
2nd week	238.5 ± 18.1	243.1 ± 12.2	221.0 ± 5.3	240.9 ± 15.5
3th week	367.4 ± 17.9	308.8 ± 17.9 *	341.5 ± 10.6	336.7 ± 2.8
Total gestational weight gain	150.9 ± 23.6	114.1 ± 10.2	153.7 ± 10	127.8 ± 4.8
Litter size (mean ± SEM)	13.0 ± 0.7	13.0 ± 0.8	13.1 ± 0.7	13.1 ± 0.9

^#^*p* < 0.01, significantly different from the 1st day; ° *p* < 0.01, significantly different from the 2nd week; * *p* < 0.05, significantly different from the control at the same week.

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
