# Peer review of "Prenatal Nutrition Containing Bisphenol A Affects Placenta Glucose Transfer: Evidence in Rats and Human Trophoblast"

_nutrients, 2020, doi:10.3390/nu12051375_

Round 1
Reviewer 1 Report
Review Report
- A summary
The study was aimed at investigating the effects of Bisphenol A (BPA) on placenta glucose transfer (GLUT) in vivo in rats and in vitro using cell cultures of human trophoblast, HTR8-SV/neo cells. Some pregnancy outcomes such as maternal weight, fetal weight, and placental weight are also evaluated in the in vivo study.
- Broad comments
The experiments are logical, seemed to be carefully done and the paper is thoroughly written, and the results were systematically presented and discussed. However, some reformulations are needed in the discussion part, especially in citation and comparison of the results from this current study and more relevant previous studies.
- Specific comments
General comment about the introduction:
To clarify the motivation for the work, I would suggest mentioning why different types of approaches (in vivo in rats and in vitro in humans) are used in this current study. What is the practical relevance to investigate these different approaches and how can one use the results from rats to extrapolate placental toxicity in humans? It is necessary to mention it in the introduction and discuss it in the discussion part. The extrapolation of placental toxicity from rats to humans should be done cautiously when one will evaluate the risk factors. Although rat and human placenta are anatomically classified as discoid and hemochorial types, there are some differences between rats and humans in terms of the placental histological structure, the fetal-maternal interface, and the developmental features.
If the reason to have two different approaches in the study is not link the results from in vivo to in vitro studies, you should clearly explain why you choose such study designs.
Page 1 line 13: Should be replaced by die with day? (also, in the whole manuscripts?)
Page 1 line 15: The abbreviation “GLUT1” should be explained as whole since it is used for first time in the text.
Page 1 lines 29-30: Provide a relevant citation.
Page 1 line 35: The abbreviation “SLC2” should be explained as whole since it is used for first time in the text.
Page 1 lines 35-37: Provide some previous study to support this statement.
Page 2 line 57: Reference nr. 18 did not investigate the effects of BSA, but para-Nonylphenol. I would suggest that you delete or replace it with another relevant study.
Page 2 line 85: Why did you not measure daily weight or some more measurement during each week to could evaluate weight gain continuously?
Page 2 line 86: There is a reduction in number of individuals in some of treatment groups. Please mention the reason for this reduction.
Page 3 line 107: The abbreviation “DMEM” should be explained as whole since it is used for first time in the text.
Page 3 line 117: The abbreviation “MPER” should be explained as whole since it is used for first time in the text.
Page 4 line 168: Which correlation test (between FW & PW) did you performed and how did you test the assumptions?
Page 4 line 172: Mention the company that developed Image J software in the current study.
Page 4 line 181: Table 2: Did you measure the initial weight of the mothers at the beginning of the study?
Page 5 line 192: Table 1: Did you find any significant differences in total gestational BW between control (150.9 ± 23,6) and BPA 2.5 (114,1 ± 10.2)?
Page 6 line 204: Figure 1:
- Please mention in the caption what the red lines are in the figures (a-c).
- How did you deal with the outliers and the variation especially in Fig. 1a (Ct and BPA 2.5)?
- The results are reported based on individual data. Did you consider/analyze the litter results as well? How were the effects when you took the litter results in the data analysis?
Page 10 line 295: General comment about the discussion
The discussion part consists of a lot of results that are previously mentioned in the results part. It has been used a lot of references and citations. They are not effectively related to your finding and discussing the results from this current study. I would suggest reducing the number of citations/references in this part and try to argue and discuss your findings more effectively with some relevant previous studies.
Page 11 line 318: I would suggest that you cite a previous study to discuss/compare your results with.
Page 11 line 337: Do you have any explanation for this finding? Are there any previous studies that are reported? I would suggest including such an explanation. That BPA is acting differently in different concentrations and that the weak dose-related responses are reported is a very interesting finding that could be immersed.
Page 11 line 340: Please discuss how the results in your study are related to this cited reference (42). I would suggest a reformulation to clarify the message.
Page 11 line 341: It will be reasonable to discuss also the effects of BPA on placenta morphology and intrauterine growth restriction (IUGR) in the different stages of gestation. You can discuss your results that are presented the changes in approximately full-term gestation (late gestation) with some other previous studies that performed the exposure study in early pregnancy.
Page 11 line 344 and page 12 line 358: Compare your results with a cited study (21) that showed a significant increase in levels of Glut-1 mRNA(197% of control; p < 0.05).
Page 12 line 379: You should mention this finding with a certain caution since the effectiveness of doses in toxicological studies should be addressed when one performs a toxicokinetic study along as clinical or molecular evaluations. In addition, the exposure method in your study is orally through drinking water. The exposure via contaminated food may differ from exposure with pure toxin through drinking water.
Author Response
We thank the reviewer for the comments. We found the suggestions useful for improving the job. We have answered all the points raised, as it can be seen in the attached file.

Reviewer 2 Report
This study aimed to assess the effects of BPA exposure on GLUT1 expression in the placenta. To complete this analysis the authors have used a rat model of dietary exposure to BPA before and during pregnancy, as well as an in vitro model with an immortalized human trophoblast cell line. For the rat studies, the authors also correlate some of their findings to maternal weight gain and fetal/placental weight at term. While this is a well-constructed study, there are some issues with the presentation of the data and conclusions that should be addressed.
Abstract
- would make more sense to read placental glucose transfer and pregnancy outcomes.
- What is ug/Kg/die? Is it Kg of diet? If so I would add the ‘t’ for clarity
Introduction
- Overall, this is nice and concise. However, the extended commentary around GLUT transporters is a little out of place. It would be better to provide more detail on the known (or unknown but rationalised) relationship between BPA and GLUT1. For example, is it known/thought to alter transcription? Transmembrane location etc?
Materials and methods
- Animal numbers presented in sections 2.1 and 2.2 do not match. It appears animals were lost between supplementation and tissue collection. The authors need to comment on what happened to these rats.
- I don’t really understand how BPA doses were administered as ug/kg/die when BPA was added to the drinking water and this presumably being provided ab lib.
- Please clearly state from when in gestation the 20 days of BPA exposure commenced and ended. What was the total length of exposure based on a standard rat gestation? (0.8?). Not all readers will necessarily be familiar with rat gestation.
- Were weights checked or recorded? Please consider the correct terminology.
- Please be specific about how many placentas were collected from each dam in each treatment group for both western and IHC analysis? Were these randomly selected or were the same positions along the uterus collected? This is important as weights likely differ along the horn, based on location. Did you ensure equal numbers of male and female pups were included in each group/analysis?
- How were the placentas sectioned? Did you ensure a full face (including lab and SZ) were visible for each specimen?
- Was the GLUT1 primary antibody specific to rats? Omission of the primary is not a sufficient negative control. You should use an appropriate IgG or antibody pre-absorption to confirm there is no non-specific binding of the primary.
- Please provide more detail on how sections were imagined and GLUT1 staining density determined? Including whether the researcher was blinded to treatment group.
- Please provide detail on the incubation conditions (O2, CO2, N2 tensions) for the HTR-8/SVneo cultures.
- Please provide detail of positive and negative controls for western blot analysis. How many samples from each treatment group were included in the analysis? How was the data normalised across multiple membranes? What software was used to determine the ODs of each band?
- Can you please provide the full name of ATP1A1 in section 2.8. It is unclear why this protein has been introduced to the analysis at this point. Were data analysed as OD? Area coverage? Please provide more detail.
- For the in vitro studies, why not compare all groups using an ANOVA to determine the effects of the different doses of BPA more thoroughly?
Results
- Line 187, what do you mean by inefficacious? Do you mean this dose had no effect?
- Comparing raw fetal weights (figure 1) does not account for variations in litter size, and distribution along the uterine horns, which are known to effect both fetal and placental weights. I know you show in table 1 no differences, but it is more conventional to present these types of data as an average per litter (i.e. n=5-8/treatment). Can you make any comment about fetal sex distribution?
- There is no reference to the correlations in the statistical analysis section. Please include detail on what test you used.
- I would remove line 210. This is a discussion point that does not belong in the reporting of the results.
- Can figure 2B be displayed as a scatter plot (is in figure 1) so that the variation in expression and exact number of samples included in the analysis per treatment group can be determined? This comment also applies to graphs in figure 4.
- IHC paragraph currently reads like a figure legend. You should use the text to describe the findings, not the figure you have included to support the findings.
- Did you quantify the IHC data? If not why not? And if not you need to make it clear that these are qualitative observations.
- Line 253 is too bold. The use of an immortalized human trophoblast cell line does not equal what is happening in the human placenta.
- It may be the pdf, but the staining patterns in figure 5A are hard to see. You may need to adjust the brightness to make this clearer
Discussion
- line 296, concluding that BPA had effects on gestational weight gain when one dose lead to a change at one time point (and not at the end of gestation) is misleading.
- Line 327, the word functional is repeated.
- The discussion could be improved by discussing how the evidence generated using the in vitro and in vivo experiments validate your findings. More discussion on why the lowest dose produced the most robust response (and the likely mechanisms behind this) would be good.
Conclusion
- I think the use of the word 'profound' is misleading. Your results show a moderate impact on GLUT1 expression patterns and changing to fetal growth.
- Be careful, you didn’t measure glucose transfer directly (line 371).
- Some of the conclusions are overreaching based on the data generated by this study. I would suggest rewording to detail the future studies that should now be completed to build upon your initial findings, which may ultimately lead to recommendations about the use of products containing BPA by women wishing to have a baby.
Author Response

(The authors gave the same response as above.)

Reviewer 3 Report
I appreciate the opportunity to review your study. I thought the work was well designed, well controlled and well presented. Excellent job.
I have only minor comments; please consider the following. The authors utilized three doses of BPA with effects generally present at the lowest dose, absent at the mid-level dose, and for some assays present at the highest dose. Can the authors speak to the biphasic nature of BPA responses and do they feel that their data are a demonstration of this biphasic nature? Additionally, can the authors provide a bit more detail about the usefulness of the immortalized placental cell line to model, without other cell types present in the experiment, the in-vivo effects of BPA at the placental.
Author Response
We thank the reviewer for the laudatory comments. We have answered to the points raised, they made us very happy because we really believed in this study and we did it with great enthusiasm.
We have answered to the points raised, as it can be seen in the attached file.

Round 2
Reviewer 2 Report
Thank you for thoroughly addressing my initial comments. I am satisfied with the adjustments made to the manuscript.